# The Emerging *Fusarium graminearum* NA3 Population Produces High Levels of Mycotoxins in Wheat and Barley

**DOI:** 10.3390/toxins16090408

**Published:** 2024-09-20

**Authors:** Nicholas A. Rhoades, Susan P. McCormick, Martha M. Vaughan, Guixia Hao

**Affiliations:** 1USDA, Agricultural Research Service, National Center for Agricultural Utilization Research, Mycotoxin Prevention and Applied Microbiology Research Unit, 1815 N. University, Peoria, IL 61604, USA; nicholas.rhoades@usda.gov (N.A.R.); susan.mccormick@usda.gov (S.P.M.); martha.vaughan@usda.gov (M.M.V.); 2Oak Ridge Institute for Science and Education, USDA, Agricultural Research Service, National Center for Agricultural Utilization Research, Mycotoxin Prevention and Applied Microbiology Research Unit, Peoria, IL 61604, USA

**Keywords:** *Fusarium graminearum*, mycotoxin, fungal growth, population, chemotype, trichothecene, Fusarium head blight

## Abstract

*Fusarium graminearum* (*Fg*) is the primary causal agent of Fusarium head blight (FHB) in wheat, barley, and other small grains in North America and worldwide. FHB results in yield reduction and contaminates grain with mycotoxins that pose threats to human and livestock health. Three genetically distinct North American (NA) populations of *Fg* have been characterized, which are generally associated with differences in their predominant trichothecene chemotype: NA1/15-acetyl-deoxynivalenol (15-ADON), NA2/3-acetyl-deoxynivalenol (3-ADON), and NA3/3α-acetoxy, 7,15-dihydroxy-12,13-epoxytrichothec-9-ene (NX-2). Recent studies found that the NA3 population had significantly less spread on point-inoculated wheat spikes than the NA1 and NA2 populations, and NX toxins are important for *Fg* spread and initial infection in wheat. In this follow-up study, to compare the effect of the three populations on initial infection and mycotoxin production on different hosts, we dip-inoculated spikes of the moderately resistant wheat cultivar Alsen and the susceptible barley cultivar Voyager using five strains from each population to evaluate disease, trichothecene mycotoxin accumulation, and trichothecene production per unit of fungal biomass. In dip-inoculated wheat spikes, the NA3 population produced significantly more trichothecene per unit of fungal biomass and accumulated higher levels of trichothecene per plant biomass than the NA1 and NA2 populations, regardless of the disease levels caused by the three populations. In contrast to its critical role during wheat infection, NX toxins had no effect on barley infection. In dip-inoculated barley, the NA1 population was more infectious and caused more severe FHB symptoms than the NA2 and NA3 populations; however, the NA3 population produced significantly higher toxin per unit of fungal biomass in infected barley tissues than the NA1 population. This study provides critical information on the emerging NA3 population, which produces high levels of NX toxin and poses a potential food safety concern.

## 1. Introduction

Fusarium head blight (FHB) is an economically and agriculturally important disease of wheat, barley, and other small cereal crops [1]. In addition to the economic losses associated with FHB, diseased grains are contaminated with harmful mycotoxins, which threaten the safety of food and feed [2]. *Fusarium graminearum* (*Fg*) is the primary source of FHB in North America and produces a variety of mycotoxins to facilitate the spread of disease within a wheat spike [3,4,5]. For instance, deoxynivalenol (DON) functions as a virulence factor, facilitating disease spread through the infected wheat spike [6], whereas NX toxins are important for *Fg* spread and initial infection in wheat spikes [7]. 

Trichothecenes inhibit eukaryotic protein synthesis, and exposure to these toxins is associated with immune, reproductive, and gastrointestinal disorders in animals [8,9,10,11]. Trichothecenes are characterized by a tricyclic 12, 13-epoxytrichothe-9-ene structure and can be classified into two main categories based on the residue at the C8 position [12]. Type A trichothecenes have an ester, a hydroxyl group, or no residue at the C8 position, while Type B trichothecenes are characterized by a keto group at C8 [13]. *Fg* isolates in North America primarily produce Type B trichothecenes, such as DON and its derivatives, 15-acetyl-DON (15-ADON) and 3-acetyl-DON (3-ADON) [14,15,16]. Nivalenol (NIV), another known Type B trichothecene, is not commonly associated with wheat in North America, but is more prevalent in corn [17]. Historically, *Fg* was not known to produce Type A trichothecenes; however, recent reports have identified an emerging North American population of *Fg* that produces a Type A trichothecene called 3α-acetoxy, 7,15-dihydroxy-12,13-epoxytrichothec-9-ene (NX-2) [18,19,20]. 

Trichothecene synthesis in *Fusarium* species is catalyzed by the *TRI* cluster, a biosynthetic gene cluster located on chromosome 2 of *Fg* [13]. The first step of trichothecene synthesis is conserved between Type A and Type B trichothecenes, with Tri5, a trichodiene synthase, performing the first committed enzymatic step in the synthesis of both groups. In the absence of *TRI5*, the synthesis of both Type A and Type B trichothecenes is abolished [7,13,21]. In *Fg*, alleles of *TRI8*, a trichothecene esterase, deacetylate di-ADON at either position C-3 or C-15 to determine whether 3-ADON or 15-ADON trichothecene analogs are produced [13,22]. In 15-ADON and 3-ADON producing strains, *TRI1* encodes a cytochrome P450 monooxygenase that facilitates the hydroxylation of positions C7 and C8 [23]. However, in NX-2 producing strains, Tri1 only hydroxylates position C7, resulting in the lack of a keto group at C8 and producing the Type A trichothecene [20]. In all chemotype and genetic backgrounds, *TRI1* is located outside of the *TRI* cluster, on chromosome 1. 

Based on genetic and genomic studies, *Fg* strains in North America (NA) associated with FHB have been grouped into three populations, named NA1, NA2, and NA3, which predominantly produce 15-ADON, 3-ADON, and NX-2, respectively. Many studies have compared the aggressiveness of the NA1 and NA2 populations over the last decade [3,4,5,24]. The NA3 population is the most recently identified *Fg* population in North America and predominantly produces the Type A trichothecene NX-2 in vitro [3,4,25,26]. NX-2 producing *Fg* strains were originally identified as low-abundance endophytes in wild grasses and were only later found to cause FHB in cultivated wheat [20]. Like 15-ADON and 3-ADON being converted to DON, the majority of NX-2 is deacetylated to NX-3 in planta. Collectively, NX-2 and NX-3 toxins can be referred to as NX toxins. NX-3 has comparable toxicity to DON in planta and in animals [7,9,27,28]. 

In general, FHB resistance in wheat and barley can be classified into two types. Resistance to initial infection is considered type 1 resistance. Initial infection level is measured by the number or percentage of florets displaying symptoms following whole spike inoculation via spray or dip method [29]. Both wheat and barley lack strong type 1 resistance. The wheat cultivar (cv) Alsen is a hard red spring wheat (*Triticum aestivum* L.) that contains Fhb1 and Fhb5 traits and exhibits a moderate level of resistance to FHB spread and initial infection [30]. Resistance to FHB spread is considered type 2 resistance and is evaluated by single floret point inoculation. A recent study investigating the differences in type 2 resistance in wheat found that the NA3 population was less aggressive but did not produce less toxin in wheat cultivars than NA1 and NA2 counterparts [31]. Like DON, NX-3 trichothecene is required for *Fg* spread in wheat spikes; additionally, NX-3 toxin is also important for initial infection in wheat [7]. We hypothesize that the NA3 population produces more toxin to enhance its infection ability in wheat. To support this hypothesis, we selected five strains from each population corresponding to their respective dominant chemotypes (15-ADON, 3-ADON, and NX-2) and investigated the potential differences in initial infection between the three populations and associated toxin production in wheat. Compared to wheat, barley has stronger type 2 resistance, and DON does not function as a virulence factor in this host [32,33,34]. Therefore, to determine whether the NA3 population has different infection behavior on different hosts, we also examined the role of NX in barley infection and compared the virulence of the three *Fg* populations on barley cv. Voyager, a two-row spring barley susceptible to FHB. 

## 2. Results

### 2.1. No Effect of Fg Populations on Initial Infection in Wheat cv. Alsen

To determine whether there were any population-dependent differences in the initial infection of wheat, whole-spike dip assays were performed on the moderately resistant wheat cv. Alsen. Although efforts were made to control the inoculation time and growth chamber conditions, it was determined that the three experimental replicates were statistically different (*p* < 0.001); therefore, the data were analyzed by experiment and depicted separately (Figure 1A–C). For experiment A, disease levels were compared at 4 and 7 dpi, and no significant difference was observed (Figure 1A). Relative fungal biomass in infected tissues, which more accurately represents fungal colonization and growth, was also compared. No significant differences in fungal biomass were observed in infected wheat tissues at 7 dpi (Figure 1A). Due to the aggressiveness of NA2 strains [35,36], there may be some levels of disease spread by NA2 strains in wheat at 7 dpi, which could not be separated from the initial infection. Therefore, disease levels and fungal biomass were only compared at 4 dpi in experiments B and C. No consistent trend in disease levels was observed at 4 dpi between experimental replicates (Figure 1B,C). No statistically significant differences in relative biomass were detected at 4 dpi (Figure 1B,C). These results suggest that there are no population-dependent differences in initial infection in wheat cv. Alsen (Figure 1A–C). 

### 2.2. Significantly Higher Levels of NX-3 Toxin Detected in Wheat Spikes during Initial Infection 

Next, we compared toxin accumulation in infected tissues and toxin content per unit of fungal biomass. The NA3 population produced significantly higher levels of toxin (toxin per fungal biomass), and wheat spikes infected with NA3 population strains accumulated significantly higher amounts of toxin in infected tissues in all three experimental replicates (Figure 2A–C). For experiment A, toxin data were collected at 7 dpi instead of 4 dpi; however, this did not appear to affect the observed trend (Figure 2A–C). These data suggest that the NA3 population produces and accumulates more toxin in wheat cv. Alsen than the NA1 and NA2 populations during initial infection. 

### 2.3. NX Toxin Is Not Involved in Barley Infection

Since NX toxins are important for initial infection in wheat [7], we examined whether they play a role in barley infection. The barley cv. Voyager was dip inoculated with the NA3 strain 44211 and its mutants 44211Δ*tri5*-3 and -19. No difference in disease level was observed between barley spikes inoculated with 44211 in comparison to the Δ*tri5* mutants (Figure 3A). As expected, NX-3 was detected in 44211 inoculated barley spikes, whereas no NX-3 was detected in Δ*tri5* mutant-inoculated barley spikes (Figure 3B). This suggests that NX is not involved in the infection of barley. 

### 2.4. NA1 Population Displays High Virulence during Infection of the Barley cv. Voyager 

Whole-spike dip assays were performed on Voyager to determine whether any population-dependent effects exist in this host. Similar to the experimental replicates in wheat, the two barley experimental replicates could not be combined statistically, so the data were analyzed and presented independently (Figure 4A,B). In contrast to wheat, the NA1 population caused significantly higher disease levels in barley cv. Voyager than the NA2 or NA3 populations did at 4, 7, and 10 dpi (Figure 4A,B). The NA1 population also had significantly higher relative biomass than the NA3 population in experiment A (Figure 4A, NA1 vs. NA2, *p* = 0.004; NA1 vs. NA3, *p* = 0.001). These results indicate that the NA1 population is more virulent on barley cv. Voyager than the NA2 and NA3 populations.

### 2.5. NA3 Population Produces Higher Levels of Toxin per Unit of Fungal Biomass in Barley Compared to NA1 and NA2 Populations

Although the NA1 population caused significantly higher disease levels in barley cv. Voyager than the NA2 or NA3 populations, the NA3 population produced higher toxin levels per unit of fungal biomass than the NA1 and NA2 populations in experiment A (Figure 5A, NA3 vs. NA1, *p* < 0.001; NA3 vs. NA2, *p* = 0.003), but only significantly higher than the NA1 population in experiment B (Figure 5B, NA3 vs. NA1, *p* = 0.003; NA3 vs. NA2, *p* = 0.118). The NA3 population also produced higher levels of toxin per gram of barley tissue than the NA1 and NA2 populations, however these differences were not statistically significant (Figure 5A,B). These data indicate that the NA3 population produces more toxin per fungal biomass than NA1 in barley cv. Voyager. 

## 3. Discussion

This study is the first to demonstrate that the emerging NA3 population is more toxigenic during the early stages of wheat and barley infection. Although the experiments were conducted on only one wheat cv. Alsen and one barley cv. Voyager, our findings showed that the NA3 population produced higher levels of toxins in both plant species. Further studies are needed to test more wheat and barley cultivars, and field surveys are also needed to confirm our observations. Previous work found that the NA3 population caused less FHB spread than the NA1 and NA2 populations on wheat spikes [31]. In the current study, we found that the less aggressive NA3 strains caused overall similar levels of initial infection on wheat as NA1 and NA2 strains. However, the disease levels caused by the three populations varied between experiments (Figure 1). Variability between experimental replicates is common in FHB disease assays due to their high sensitivity to environmental conditions, which can fluctuate between experimental replicates initiated at different times [29]. Infection was initiated when wheat spikes were at mid-anthesis (around 30–70%) and barley plants were heading; however, the development of spikes within a growth chamber can be asynchronous, which may also contribute to some experimental variability. Regardless of the levels of disease variation, our results showed that the NA3 population produced significantly greater toxin per unit of fungal biomass and accumulated more toxin per plant biomass during early-stage infection of wheat (Figure 2). Similar to DON, NX-3 acts as a virulence factor in wheat to facilitate fungal spread through the rachis nodes and colonize the whole spike [6,7]. In contrast to DON, which is not critical for initial infection [33], NX-3 plays an important role in initial infection of wheat, with NX-3-deficient strains showing reduced initial infection compared to their wild-type parent [7]. It has been suggested that the NA3 population may have recently switched from an endophytic lifestyle in wild grasses to being a pathogen on domesticated cereal grains [25,40]. We speculate that the NA3 population may have evolved to produce more toxin to enhance its ability to infect hosts that it may not be fully adapted to, such as wheat and barley. This may partially explain why we observed that the NA3 population produced and accumulated higher amounts of NX toxin in wheat. On the other hand, NA3 population strains may produce more toxin on wheat and barley to better compete with the established NA1 and NA2 populations. Further experiments are needed to test the role of NX-3 in inter-population competition on different hosts.

The NA1 population was more infectious than NA2 and NA3 in barley cv. Voyager; however, the NA3 population produced more toxin per unit of fungal biomass than NA1. This may be a consequence of the NA3 population growing less than NA1, as determined by fungal biomass. NA3 fungal biomass was significantly lower than NA1 and NA2 in experiment A (Figure 4, NA1 vs. NA2, *p* = 0.004; NA1 vs. NA3, *p* = 0.001), and the biomass of NA3 was 60% of NA1 biomass in experiment B, but the difference was not significant (Figure 4, NA1 vs. NA2, *p* = 0.106; NA1 vs. NA3, *p* = 0.275). 

In addition to trichothecene analog differences between populations, several population-specific virulence factors, such as effector proteins, have been identified in *Fg* that influence its pathogenic potential [26]. Kelly and Ward found that gene flow was apparent between the three North American populations; however, each population retains unique sets of genes, some presumed to be involved in virulence [26]. Taken together, this suggests that trichothecene-analog production is not likely to be the sole determinant of pathogenicity. Variations in population-specific genes, such as effectors, are likely involved in determining how aggressive a population may be on any particular host [26,41,42]. 

With the occurrence of genetic recombination between populations expected to increase over time, it is important to identify the dynamics of how the NA3 population interacts with domesticated cereal crops [26,31]. NX-3 toxin has similar toxicity to DON in Arabidopsis and animal systems [43], and our results showed that NX-3 accumulates at higher levels per unit of fungal biomass than DON in both wheat and barley tissues. This study shows that it is necessary to survey and screen the presence of NX contamination in wheat and barley grain and commodities to assess potential risk. Furthermore, understanding how disease and toxin production are affected by mixed populations may help improve future disease prediction models.

## 4. Conclusions

In summary, the three North American *Fg* populations caused similar levels of initial infection in wheat, but the emerging NA3 population produced and accumulated more mycotoxins in infected spikes than the NA1 and NA2 populations. In barley, the NA1 population caused more visual symptoms of FHB, but the NA3 population produced more toxin per unit of fungal biomass than the NA1 population. Despite the increased toxin production, we determined that NX toxins do not enhance infection of barley by NA3 population strains. The discovery that NA3 population strains can produce more NX in wheat and barley and accumulate more NX in wheat emphasizes the need to assess the frequency and level of NX contamination in cereal crops.

## 5. Materials and Methods

### 5.1. Fusarium Strains and Cultivation

Five strains from each population (NA1, NA2, and NA3) used in this study are listed in Table 1. Each strain was previously characterized for genotype using SNP analysis [26]. The chemotype of each strain was previously characterized using either multi-locus genotyping or gas chromatography-mass spectrophotometry (GC-MS) screening [31]. All strains were maintained on V8-agar (V8: 113 mL V8 juice, 1.75 g CaCO_3_, 11.7 g agar, 467 mL H_2_O) with a 12:12 h light/dark cycle at 28 °C. 

NA3 strain 44211 *TRI5* deletion mutants (44211∆*tri5*-3 and -19), which were generated in our previous studies by one-step construction of *Agrobacterium*-recombination-ready-plasmids (pOSCAR) and *Agrobacterium*-mediated transformation [44], were maintained on V8-agar with a 12:12 h light/dark cycle at 28 °C, as described above.

### 5.2. Cultivation of Wheat and Barley 

The wheat cv. Alsen, moderately resistant to FHB, was planted in 7-inch pots. Five seeds were sown in SunShine Mix (Sun Gro Horticulture, Agawam, MA, USA) with the addition of 100 g Osmocote and 15 g Micromax in 5 L soil. The plants were maintained in a controlled growth house with 16 h of light at 23 °C and 8 h of dark at 20 °C with 50% relative humidity. Wheat plants were watered daily and fertilized every two weeks with a solution containing 325 mg/L of Peter’s 20:20:20 (Grace-Sierra Horticultural Products, Milpitas, CA, USA) until inoculation.

The barley cv. Voyager, susceptible to FHB, was used in the disease assays. Four plants were grown in Miracle-Gro/Moisture Control soil (Miracle-Grow, Marysville, OH, USA) mixed with 100 g Osmocote containing micronutrients and maintained in similar growth conditions as wheat.

### 5.3. FHB Virulence Assays 

For evaluation of initial infection, whole-spike dip inoculation was performed as described [7]. Briefly, to prepare macroconidia for wheat and barley infection assays, three V8-agar discs containing mycelial growth were transferred to 20 mL of mung bean broth and cultured for 4 days at 28 °C with shaking at 200 rpm [45]. Mung bean cultures were filtered through a 40 µm cell strainer (Biologix, Jinan, China) and centrifuged for 10 min at 3000 rpm. The pellet was washed with water and adjusted to a concentration of 10^4^ conidia/mL for wheat inoculation and 10^5^ conidia/mL for barley inoculation in 50 mL of 0.02% Tween made with purified DI-water (Thermo Fisher Scientific, Waltham, MA, USA). Disease severity was evaluated by counting the number of diseased spikelets and dividing it by the total number of spikelets per spike. 

Eighteen wheat or barley spikes per strain were submerged in the macroconidia suspension at mid-anthesis. Inoculated wheat or barley spikes were covered with a plastic bag and sealed for three days to maintain high humidity. FHB progress in wheat was evaluated at 4 days post inoculation (dpi), except for the first replicate of the Alsen trials, in which FHB progress was evaluated at 4 and 7 dpi. After 7 dpi, FHB can spread in Alsen wheat spikes inoculated with some aggressive strains [31]. In contrast, barley cv. Voyager has strong type 2 resistance, which restricts FHB symptom spread beyond the initially infected spikelet. Therefore, FHB progress in barley was evaluated at 4, 7, and 10 dpi. At the final experimental timepoint, the inoculated spikes were harvested, lyophilized, and ground for toxin extraction and *Fg* biomass estimation. Three experiments were performed for wheat, due to the first replicate being harvested at 7 dpi rather than 4 dpi. Experiments were performed in duplicate for barley assays.

### 5.4. Fungal Biomass Quantification

Fungal biomass in the inoculated tissues was estimated as the ratio of *Fg* DNA/host DNA using quantitative PCR (qPCR) as previously described in Laraba et al., 2023. Briefly, infected wheat or barley spikes were lyophilized and then pulverized with a GenoGrinder. Genomic DNA was extracted from ~50 mg of lyophilized and ground tissues using the ZR Fungal/Bacterial DNA Miniprep Kit (Zymo Research, Boston, MD, USA). The qPCR reactions were performed using the Fluidigm Biomark HD/Juno system (South San Francisco, CA, USA). Species-specific primers and probes used are listed in Appendix A. Other PCR components, thermal cycling conditions and fluorescence detection were carried out following the manufacturer’s protocol 192.24 IFC for Gene Expression (Fluidigm, South San Francisco, CA, USA). The resulting raw fluorescence data were then used to estimate PCR efficiency per sample and to calculate the initial DNA concentration using the LinReg PCR data analysis program. 

### 5.5. Trichothecene Detection in Planta

Ground tissue (about 0.5 g) was extracted in 50 mL Falcon tubes with 10 mL acetonitrile/water (86:14). After centrifugation, 5 mL of the extract was purified with a Romer MycoSep cleanup column. Two mL of the purified extract was dried in a 1-dram vial with heat under a stream of air. Trimethylsilyl (TMS) derivatives were prepared by adding 100 μL of a 100:1 freshly prepared mixture of N-trimethylsilylimidazole/trimethylchlorosilane to the dried extracts. After a 30 min reaction time, 900 µL of isooctane was added to the vial and briefly vortexed to mix. One mL of water was then added to the vial to quench the reaction, and it was gently mixed until the top layer became clear. The top organic layer was transferred to a GC vial for gas chromatography-mass spectrometry (GC-MS) analysis.

GC-MS analyses were performed with an Agilent 7890 chromatograph (Wilmington, DE, USA) fitted with an HP-5MS column and an Agilent 5977 mass spectrometer with an electron impact source operating in selected ion monitoring (SIM) mode. Samples were introduced with splitless injection at 150 °C, the temperature was held for 1 min, then the column was heated at 30 °C/min to 280 °C and held for 1 min. Under these conditions, 3,7,15-tri-trimethylsilyl DON is detected at 6.2 min using ions 512, 422, 392, 295, 259, and 235 and 3,7,15-tri-trimethylsilyl NX-3 is detected at 6.8 min using ions 408, 318, 305, and 181. DON and NX-3 were quantified using standard curves (0.3125 to 80 µg/mL) of tri-trimethylsilyl DON and tri-trimethylsilyl NX-3 derivatives prepared in the same way. 

NX-2 was isolated from liquid cultures of *F. graminearum* NRRL44211. NX-3 was prepared by treating NX-2 with 0.1N NaOH. DON was prepared by hydrolysis of 15-ADON isolated from liquid cultures of *F. graminearum* B4-1.

### 5.6. Statistical Analysis

All statistical analyses were conducted using JMP Statistical Discovery Software version 17 (Cary, NC, USA). To determine if data from individual experimental replicates could be combined, one-way analysis of variance (ANOVA) was performed, followed by Tukey-Kramer honestly significant differences (HSD) tests. Since experimental replicate was a significant contributing factor to differences in the data, comparisons were made by each experiment using ANOVA Tukey-Kramer HSD analyses (with a significance level of *p* < 0.05).

## Figures and Tables

**Figure 1 toxins-16-00408-f001:**
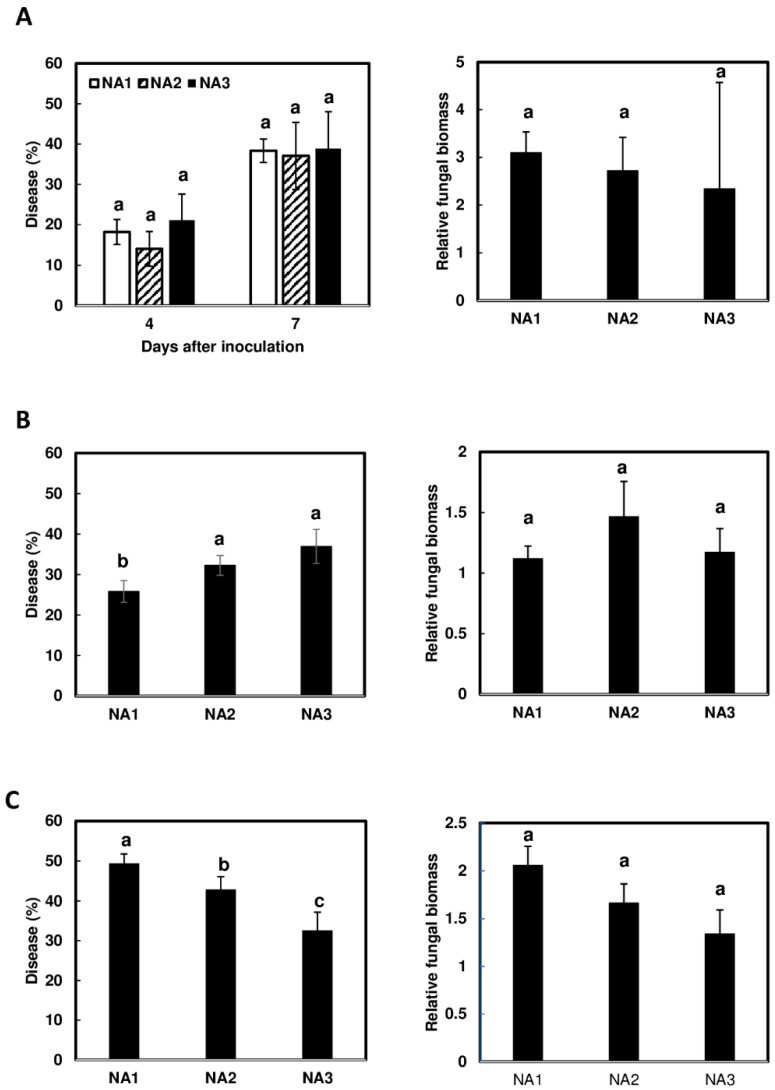
Comparison of initial infection and fungal biomass in moderately resistant wheat cv. Alsen. Alsen spikes were immersed in conidia suspensions (50 mL, 10^4^ conidia/mL) of different strains. Percentage of disease was calculated by counting the number of symptomatic spikelets (**left**), and relative biomass was determined by qPCR (**right**). Three experimental replicates are shown by independent analysis. Disease severity from experiment (**A**) was analyzed at 4 days post-infection (dpi) and 7 dpi, and biomass was determined in 7 dpi tissues. Disease severity from experiments (**B**,**C**) were analyzed at 4 dpi for disease and fungal biomass. Different letters indicate statistically significant differences based on one-way ANOVA analysis followed by Tukey-Kramer honestly significant difference (HSD) (*n* = 5; *p* < 0.05).

**Figure 2 toxins-16-00408-f002:**
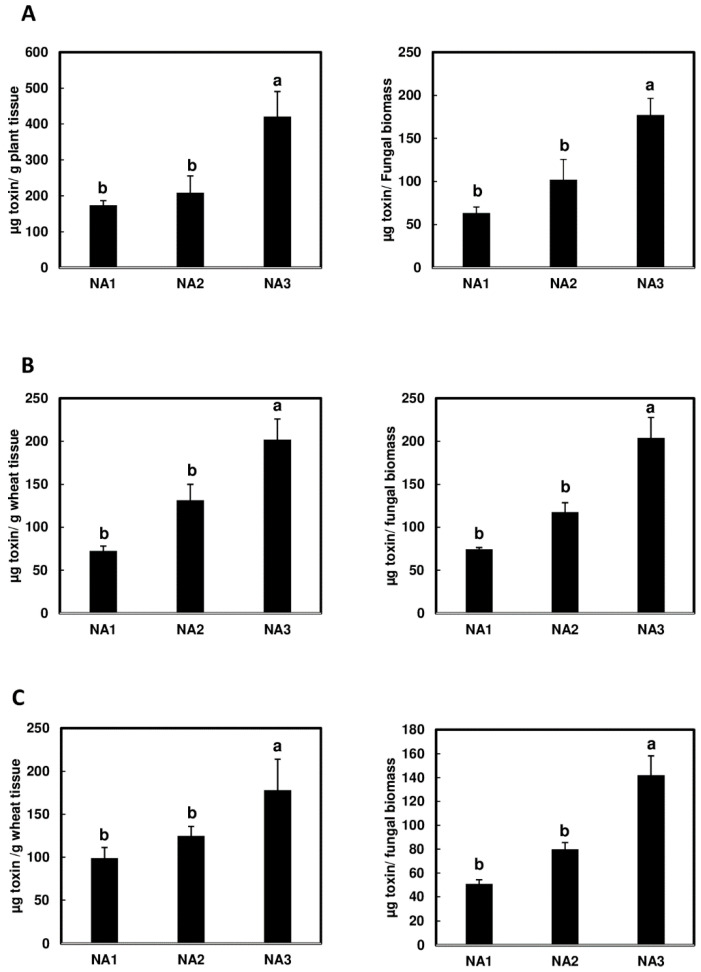
Greater amounts of trichothecene mycotoxin were detected in wheat cv. Alsen inoculated with *Fusarium graminearum* population NA3 than with NA1 and NA2. Toxin contents analyzed per plant biomass (**left**) and per fungal biomass (**right**) for three experimental replicates are shown as three independent experiments (**A**–**C**). Toxin content for experiment (**A**) was determined in 7-dpi tissues, while experiments (**B**,**C**) were analyzed at 4 dpi. Different letters indicate statistically significant differences based on one-way ANOVA analysis followed by Tukey-Kramer honestly significant difference (HSD) (*n* = 5; *p* < 0.05).

**Figure 3 toxins-16-00408-f003:**
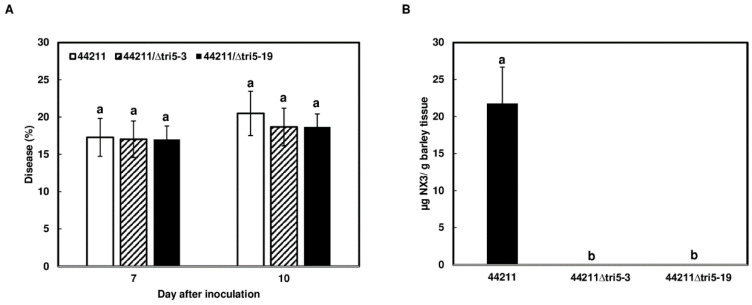
NX-3 is dispensable in the initial infection of barley. Whole spikes of barley cv. Voyager were immersed in conidia suspensions (50 mL, 10^5^ conidia/mL) of 44211 and its mutants 44211∆*tri5*-3 and -19. (**A**), FHB severity was evaluated at 7 and 10 dpi. Bars represent the mean percentages and standard errors of twelve inoculated spikes for each strain. Different letters indicate a significant difference; (**B**), NX-3 was measured from inoculated wheat spikes at 10 dpi. Bars represent the means and standard errors of NX-3 levels. Means at each time point were analyzed independently and compared using one-way ANOVA and Tukey-Kramer HSD (*n* = 12 for disease; *n* = 6 for NX-3 content). Different letters indicate significant difference at the *p* < 0.05 level.

**Figure 4 toxins-16-00408-f004:**
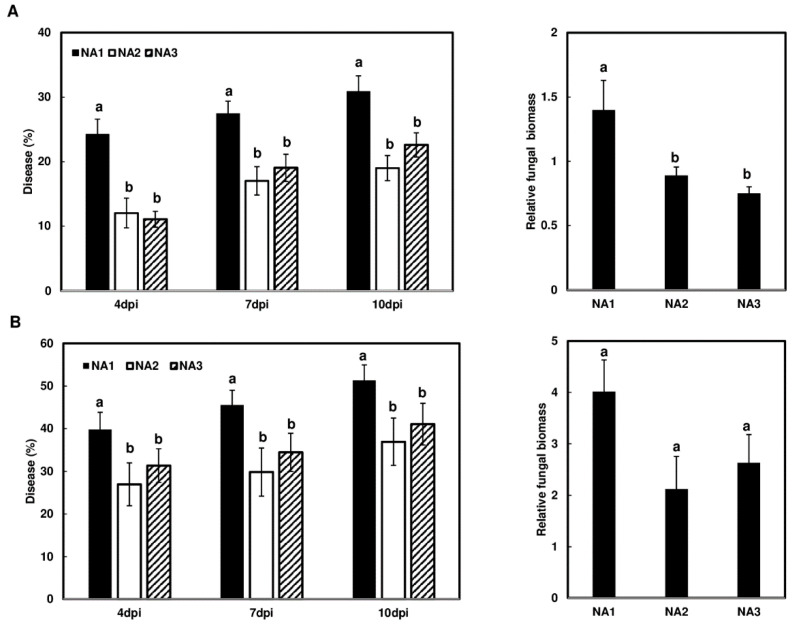
Comparison of initial infection and fungal biomass in barley cv. Voyager inoculated with different *Fusarium graminearum* populations. Whole spikes of barley cv. Voyager were immersed in conidia suspensions (50 mL, 10^5^ conidia/mL) of different strains. The percentage of disease (**left**) and relative biomass (**right**) for two experimental replicates are shown as two independent experiments (**A**,**B**). Each population (NA1, NA2, NA3) is represented by five strains (Table 1). Different letters indicate statistically significant differences based on one-way ANOVA analysis followed by Tukey-Kramer honestly significant difference (HSD) (*n* = 5; *p* < 0.05).

**Figure 5 toxins-16-00408-f005:**
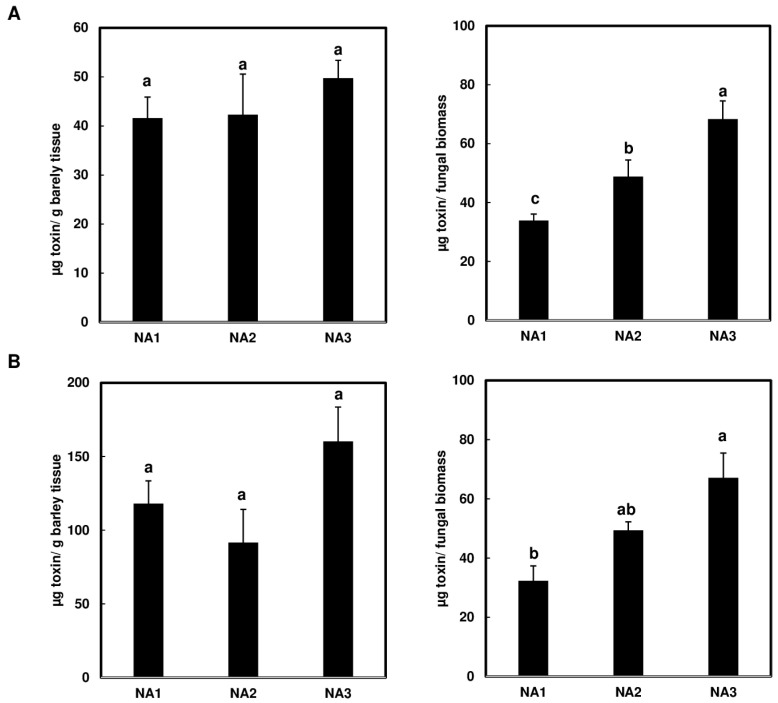
Greater amounts of trichothecene mycotoxin per fungal biomass were produced by the *Fusarium graminearum* NA3 population than NA1 and NA2 in barley cv. Voyager. Toxin contents analyzed per plant biomass (**left**) and per fungal biomass (**right**) for two experimental replicates are shown as two independent experiments (**A**,**B**). Toxin production for all experiments was analyzed at 10 dpi. Different letters indicate statistically significant differences based on one-way ANOVA analysis followed by Tukey-Kramer honestly significant difference (HSD) (*n* = 5; *p* < 0.05).

**Table 1 toxins-16-00408-t001:** Strains used in this study.

NRRL #	Alternate Name	Population	Chemotype	Isolate Location	Host	Reference
GZ3639		NA1	15ADON	Kansas (KS), USA	Wheat	[37]
PH1		NA1	15ADON	Michigan (MI), USA	Wheat	[38]
64387	13MN1-6	NA1	15ADON	Minnesota (MN), USA	Wheat	[4]
64388	F333	NA1	15ADON	North Dakota (ND), USA	Wheat	[4]
38746	ON-05-85	NA1	15ADON	Ontario (ON), Canada	Wheat	[3]
38581	Q-05-105	NA2	3ADON	Quebec (QC), Canada	Wheat	[3]
37525	S8A-04-2	NA2	3ADON	Saskatchewan (SK), Canada	Wheat	[26]
38964	ON-05-92	NA2	3ADON	Ontario (ON), Canada	Wheat	[3]
38763	Q-05-17	NA2	3ADON	Quebec (QC), Canada	Wheat	[3]
46422	00-566	NA2	3ADON	Minnesota (MN), USA	Wheat	[39]
44211	5B-06-4	NA3	NX2	Saskatchewan (SK), Canada	Wheat	[3]
43884	ON-06-4	NA3	NX2	Ontario (ON), Canada	Wheat	[3]
43161	M11-05-oat4	NA3	NX2	Manitoba (MB), Canada	Oat	[3]
66044	03-348	NA3	NX2	North Dakota (ND), USA	Wheat	[4]
64394	F322	NA3	NX2	Minnesota (MN), USA	Wheat	[4]

## Data Availability

The raw data supporting the conclusions of this article will be made available by the authors on request.

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
