# Peer review of "The Emerging Fusarium graminearum NA3 Population Produces High Levels of Mycotoxins in Wheat and Barley"

_toxins, 2024, doi:10.3390/toxins16090408_

Round 1

Reviewer 1 Report

Comments and Suggestions for Authors

This study reported that the emerging Fg NA3 is likely to produce higher level of mycotoxins in small cereal crops, which is significance of real-life production. However, the available data in this manuscript is more like the results of a pre-experiment and has not yet met the requirements of our journal for publication. Besides, the importance and novelty of this study are not well articulated the introduction part. More additional data that can soundly demonstrate higher levels of mycotoxins produced by the NA3 population are expected to be supplemented and resubmitted.

Author Response

This study reported that the emerging Fg NA3 is likely to produce higher level of mycotoxins in small cereal crops, which is significance of real-life production. However, the available data in this manuscript is more like the results of a pre-experiment and has not yet met the requirements of our journal for publication. Besides, the importance and novelty of this study are not well articulated the introduction part. More additional data that can soundly demonstrate higher levels of mycotoxins produced by the NA3 population are expected to be supplemented and resubmitted.

We agree the limitations of this manuscript. Due to space limitation, we tested total of 15 strains from three North America populations on one wheat and one barley varieties. Further investigation on more strains and more wheat and barley cultivars are planned according to our findings. In addition, field surveys are also needed to find whether our findings apply in the field. This study showed that the Fg NA3 population is a potential risk for an emerging mycotoxin contamination and more exhaustive studies are warranted. We have revised the introduction and key contributions sections to emphasize the importance and novelty of this study.

Reviewer 2 Report

Comments and Suggestions for Authors

Dear authors,

Congratulation for putting forth a very relevant study. 

This research provides valuable insights into the differential impacts of Fusarium graminearum populations on wheat and barley, with a particular focus on the emerging NA3 population. By identifying the NA3 population's unique ability to produce higher levels of mycotoxins, the study underscores the importance of continuous monitoring and tailored strategies to mitigate the risks associated with Fusarium head blight in cereal crops. The article is well written and scientifically sound. I have no comments except that conclusion could have been separately presented for more clarity. 

I wish authors all the best for future work!

Author Response

Congratulation for putting forth a very relevant study. 

This research provides valuable insights into the differential impacts of Fusarium graminearum populations on wheat and barley, with a particular focus on the emerging NA3 population. By identifying the NA3 population's unique ability to produce higher levels of mycotoxins, the study underscores the importance of continuous monitoring and tailored strategies to mitigate the risks associated with Fusarium head blight in cereal crops. The article is well written and scientifically sound. I have no comments except that conclusion could have been separately presented for more clarity. 

I wish authors all the best for future work!

Thank you very much for the nice comment!  We separated discussion and conclusion sections as suggested.

Reviewer 3 Report

Comments and Suggestions for Authors

The manuscript needs the following revisions before acceptance for publication:

(1) The use of the abbreviation (Fg) is not appropriate one, and it needs to be presented as F. graminearum

(2) Line 79-81: Check the format

(3) Introduction section alone has 37 references cited which is too high and can be reduced by avoiding related references and citing the most recent and classical papers.

(4) Line No. 212: What are the characteristics of the SunShine mix

(5) Present more details about the wheat cultivar Alsen

(6) Line No. 219: What are the characteristics of the MiracleGro/Moisture Control Soil

(7) Present more details about the barley cultivar Voyager

(8) Why are different cultures for two crops, are they crop-specific?

(9) Present the Discussion chapter with the sub-headings

Author Response

  • The use of the abbreviation (Fg) is not appropriate one, and it needs to be presented as graminearum

We are not the inventor to use this kind of abbreviation. You can find over 100 publications with similar abbreviation. A few examples: 

Laraba, et al. Insights into the aggressiveness of the emerging North American population 3 (NA3) of Fusarium graminearum. Plant Dis 2023, 107, 2687-2700, doi:10.1094/PDIS-11-22-2698-RE.

Liu et al. Hordedane diterpenoid phytoalexins restrict Fusarium graminearum infection but enhance Bipolaris sorokiniana colonization of barley roots. Mol Plant 2024 Aug 5;17(8):1307-1327.

Kimotho et al. A potent endophytic fungus Purpureocillium lilacinum YZ1 protects against Fusarium infection in field-grown wheat. New Phytol. 2024 Sep;243(5):1899-1916.

  • Line 79-81: Check the format

Corrected.

  • Introduction section alone has 37 references cited which is too high and can be reduced by avoiding related references and citing the most recent and classical papers.

As suggested, we went through the refences and deleted the following 5 references to avoid related references.  

Windels, C.E. Economic and social impacts of fusarium head blight: changing farms and rural communities in the northern great plains. Phytopathology 2000, 90, 17-21, doi:10.1094/PHYTO.2000.90.1.17.

Munkvold, G.P.; Proctor, R.H.; Moretti, A. Mycotoxin production in Fusarium according to contemporary species concepts. Annu Rev Phytopathol 2021, 59, 373-402, doi:10.1146/annurev-phyto-020620-102825.

Oghenekaro, A.O.; Oviedo-Ludena, M.A.; Serajazari, M.; Wang, X.; Henriquez, M.A.; Wenner, N.G.; Kuldau, G.A.; Navabi, A.; Kutcher, H.R.; Fernando, W.G.D. Population genetic structure and chemotype diversity of Fusarium graminearum populations from wheat in Canada and north eastern United States. Toxins 2021, 13, 180, doi:10.3390/toxins13030180.

Ward, T.J.; Clear, R.M.; Rooney, A.P.; O'Donnell, K.; Gaba, D.; Patrick, S.; Starkey, D.E.; Gilbert, J.; Geiser, D.M.; Nowicki, T.W. An adaptive evolutionary shift in Fusarium head blight pathogen populations is driving the rapid spread of more toxigenic in North America. Fungal Genetics and Biology 2008, 45, 473-484, doi:10.1016/j.fgb.2007.10.003.

Yang, J.-H.; Wang, J.-H.; Guo, W.-B.; Ling, A.R.; Luo, A.-Q.; Liu, D.; Yang, X.-L.; Zhao, Z.-H. Toxic effects and possible mechanisms of deoxynivalenol exposure on sperm and testicular damage in BALB/c mice. J Agric Food Chem 2019, 67, 2289-2295, doi:10.1021/acs.jafc.8b04783.

  • Line No. 212: What are the characteristics of the SunShine mix

The composition of SunShine mix is available on the manufacture’s website.

ï‚·  Canadian Sphagnum Peat Moss

ï‚·  Perlite

ï‚·  Dolomite Lime

ï‚·  Long-Lasting Wetting Agent

ï‚·  RESiLIENCE

  • Present more details about the wheat cultivar Alsen

Revised to: Wheat cv Alsen is a hard red spring wheat (Triticum aestivum L.) that contains Fhb1 and Fhb5 traits and exhibits moderate level of resistance to FHB spread and initial infection.

  • Line No. 219: What are the characteristics of the MiracleGro/Moisture Control Soil

We are not sure what the question is? You mean compositions of the MiracleGro/Moisture Control mix? The composition of this mix is not available on the manufacture’s website.

(7) Present more details about the barley cultivar Voyager

Added: Voyager is a two-row spring barley and is susceptible to FHB.  

(8) Why are different cultures for two crops, are they crop-specific?

I think I understand what you mean for questions 4 and 6. We usually use SunShine mix to grow wheat. The wheat plants grow healthy. However, barley plants do not grow very well in SunShine mix and we tried other soils and found barley plants grew better in MiracleGro/Moisture Control Soil. Therefore, we use this soil for barley. Hope this answered your questions.

(9) Present the Discussion chapter with the sub-headings

We have a short discussion for this manuscript, we feel it is not necessary to include sub-headings. We did separate discussion and conclusions.

Round 2

Reviewer 1 Report

Comments and Suggestions for Authors

For the results of Fig. 1, why did we detect the disease severity at 4 dpi (Fig. 1B, C)other than that at 7 dpi? There are still some minor formatting errors needing to be correct.

Author Response

For the results of Fig. 1, why did we detect the disease severity at 4 dpi (Fig. 1B, C)other than that at 7 dpi? There are still some minor formatting errors needing to be correct.

This was explained in page 3 line115-118.

Due to more aggressiveness of NA2 strains [35,36], there may be some levels of disease spread by NA2 strains in wheat at 7 dpi, which could not be separated from initial infection. Therefore, the disease levels and fungal biomass were only compared at 4 dpi in experiments B and C.  Also a few changes were made and highlighted. 
